# The Hedgehog Signaling Pathway in Ischemic Tissues

**DOI:** 10.3390/ijms20215270

**Published:** 2019-10-24

**Authors:** Igor Giarretta, Eleonora Gaetani, Margherita Bigossi, Paolo Tondi, Takayuki Asahara, Roberto Pola

**Affiliations:** 1Department of Medicine, Fondazione Policlinico Universitario A. Gemelli IRCCS, Università Cattolica del Sacro Cuore, 00168 Rome, Italy; igor.giarretta@unicatt.it (I.G.); eleonora.gaetani@unicatt.it (E.G.); margherita.bigossi01@icatt.it (M.B.);; 2Department of Regenerative Medicine Science, Tokai University School of Medicine, 143 Shimokasuya, Isehara, Kanagawa 259-1193, Japan; asa777@is.icc.u-tokai.ac.jp

**Keywords:** hedgehog, ischemia, heart, brain, skeletal muscle

## Abstract

Hedgehog (Hh) proteins are prototypical morphogens known to regulate epithelial/mesenchymal interactions during embryonic development. In addition to its pivotal role in embryogenesis, the Hh signaling pathway may be recapitulated in post-natal life in a number of physiological and pathological conditions, including ischemia. This review highlights the involvement of Hh signaling in ischemic tissue regeneration and angiogenesis, with particular attention to the heart, the brain, and the skeletal muscle. Updated information on the potential role of the Hh pathway as a therapeutic target in the ischemic condition is also presented.

## 1. Introduction

Hedgehog (Hh) signaling was first described in 1980 by Nusslein-Volhard and Wieschaus and was initially addressed as a key mediator of cellular proliferation, differentiation, and migration during organogenesis in invertebrates [1]. The general signaling mechanism for the Hh pathway is conserved from fly to mammal, although more and distinct components have been found in mammalian cells [2]. In the early 1990s, three Hh homologs were discovered in vertebrates: Sonic (Shh), Desert (Dhh), and Indian hedgehog (Ihh). Among them, Shh is the most widely expressed during development and the best studied. The initial evidence for the importance of Hh signaling in mammalian development was provided by observations that mutations in Shh cause holoprosencephaly, a developmental disorder that affects midline morphogenesis of the face and nervous system [3,4,5]. Although most studies of Hh signaling focus on Shh, the other two ligands—Dhh and Ihh—share many of the same downstream signaling components. The Dhh ligand is more abundant in the reproductive organs of both males and females, including Sertoli cells of testes and granulosa cells of ovaries [6,7]. Consistent with this, male mice lacking Dhh are infertile due to the complete absence of mature sperm [6]. Ihh instead plays an important role in skeletal development, primarily in cortical bone and long bone formation. The most severe manifestation of Ihh deficiency may be seen in children with acrocapitofemoral dysplasia, who have short stature and bone defects [8].

It was 2001 when we first reported that the Hh signaling pathway has a role in angiogenesis [9]. Before that seminal publication, there were only sparse observations pointing to the possible involvement of Hh in vascularizing certain embryonic tissues. For instance, it was known that hypervascularization of the neuroectoderm occurs upon transgenic overexpression of Shh in the dorsal neural tube [10], or that Shh-deficient zebrafish has disorganization of endothelial precursors and an inability to form the dorsal aorta or axial vein [11], or that Shh-deficient mice lack proper vascularization of the developing lung [12]. Nowadays, there is substantial evidence that the Hh signaling pathway is an important player in the regulation of angiogenesis in a number of physiological and pathological conditions in post-natal life, with multiple implications in the field of cardiovascular medicine, neurology, and oncology. This review presents an updated summary of our understanding and the many controversial issues on the role played by the Hh signaling pathway in ischemic tissues, with a specific focus on heart, brain, and skeletal muscle, which are the tissues with the most robust literature in this field.

## 2. Skeletal Muscle

### 2.1. Activation of the Hh Pathway in the Ischemic Muscle 

It is well established that ischemia induces strong upregulation of many components of the Hh pathway, including the Shh ligand, the Ptch receptor, and the transcription factors Gli1, Gli2, and Gli3, in the skeletal muscle [13,14,15,16,17]. Nonetheless, the biological mechanisms underlying this phenomenon are still not completely understood. Experimental data have shown that hypoxia per se—independently of ischemia—is able to induce a rapid systemic Hh response in various organs of adult mice, with the Hh response being preceded by the accumulation of the master transcriptional regulator of hypoxia HIF-1α (Hypoxia-Inducible Factor-1α) [18]. Pharmacological inhibition, knockdown, or genetic ablation of HIF-1α abolishes Hh pathway activation in hypoxic tissues [18]. HIF-1α inhibition is also sufficient to block Hh signaling in cancer cells [19]. Based on this, it is plausible that the activation of the Hh pathway observed in response to ischemia in the skeletal muscle may occur through HIF-1α. However, a precise role for HIF-1α, specifically in this setting, has not yet been established.

### 2.2. Effects of Endogenous Hh in the Ischemic Skeletal Muscle 

Once the Hh pathway is upregulated, it exerts important regulatory effects on angiogenesis in the ischemic muscle. There is strong evidence that such regulatory effects mainly occur in an indirect manner, with the production of angiogenic growth factors by Shh-responding cells, i.e., interstitial fibroblasts/mesenchymal cells. This has been demonstrated in vivo in a murine model of hindlimb ischemia, in which fibronectin-positive interstitial mesenchymal cells within the ischemic area produce vascular endothelial growth factor (VEGF) through an Hh-dependent mechanism [13]. The fact that the Hh pathway is an indirect stimulator of angiogenesis was originally discovered using a murine corneal model of angiogenesis, in which pellets containing Shh were implanted in the cornea of mice carrying the lacZ reporter gene under the control of the Ptch gene promoter [9]. The result was the upregulation of the lacZ reporter gene in corneal interstitial mesenchymal cells, which indicated that these cellular elements were the Shh-responding cells in this in vivo assay. It was also demonstrated that these Shh-responding cells were immunopositive for VEGF, which indicated that they were producing angiogenic growth factors in response to Shh stimulation [9]. These in vivo data were supported by a series of in vitro experiments, which demonstrated that Shh has the ability to upregulate the expression of angiogenic cytokines, including all three VEGF-1 isoforms and angiopoietins-1 and -2, in fibroblasts [9]. On the other hand, Shh has no direct effect on endothelial cell migration or proliferation [9]. The capacity of Shh to induce the production of angiogenic growth factors, including VEGF, hepatocyte growth factor (HGF), platelet-derived growth factor-BB (PDGF-BB), and insulin-like growth factor (IGF), in interstitial cells and fibroblasts, has been later confirmed by other authors in various organs and tissues [20,21,22,23]. Also, it has been confirmed that none of the Hh ligands is able to induce Gli-target genes in endothelial cells [24] and that Shh has limited effects on endothelial cell proliferation and migration [17,24]. There is also the demonstration that Gli3 deficiency in endothelial cells does not affect the repair of the ischemic skeletal muscle [25]. On the other hand, it has been reported that Hh proteins have the ability to promote endothelial cell tubulogenesis through a non-canonical Hh pathway that involves Smo, Gi proteins, and Rac1 and results in cytoskeletal rearrangement and formation of actin stress fibers in endothelial cells [24]. Nonetheless, endothelial-specific Smo knockout (eSmoNull) mice have no observable phenotype and display unaltered ability to mount an appropriate angiogenic process in response to hindlimb ischemia [26]. Even Shh-induced corneal angiogenesis is normal in eSmoNull mice [26]. Taken together, these findings support the concept that the main molecular mechanism through which the endogenous Hh pathway regulates angiogenesis in the ischemic skeletal muscle is the stimulation of growth factors production by interstitial mesenchymal cells and fibroblasts. Nonetheless, we have also demonstrated that Shh plays a role in myogenesis, as its inhibition impairs the activation of the myogenic regulatory factors Myf-5 and MyoD and reduces the number of myogenic satellite cells in the injured skeletal muscle site [27,28]. However, these myogenic effects have been demonstrated upon mechanical and toxic injury of the skeletal muscle, or in experimental models of muscle dystrophy, and it is not known whether they are relevant also in the setting of ischemia.

Some controversies exist on the identity of the Hh ligand that plays the most important role during ischemia-induced angiogenesis in the skeletal muscle. We have found that hindlimb ischemia results in upregulation of Shh expression, without a detectable effect on Ihh and Dhh [13]. We have also recently found that microparticles (MPs) carrying Shh are increased in humans with peripheral artery disease [29]. However, Caradu and coll. have reported that angiogenesis is not impaired in Shh inducible knockout mice, opening the possibility that endogenous Shh does not promote ischemia-induced angiogenesis, but has instead the ability to limit inflammation, macrophage recruitment, and chemokine expression in myocytes in the ischemic skeletal muscle [30]. This study deserves attention because it used genetic manipulation tools (Shh inducible deficient mice) to specifically inhibit the activity of endogenous Shh and test the effects of such inhibition on ischemia-induced angiogenesis. The findings of Caradu and coll. open the possibility that other Hh ligands, rather than Shh, or in addition to Shh, are important for ischemia-induced angiogenesis. Consistent with this hypothesis is the notion that Dhh is expressed in ECs and is important for proper endothelial function [31]. Also, it has been shown that limb perfusion is significantly impaired in the absence of Dhh [16]. There are data supporting the hypothesis that Ihh might also have a role in angiogenesis. In particular, Ihh is necessary for the formation of the anterior aorta and the vascularization of the yolk sac [32,33]. Ihh has also been shown to synchronize skeletal angiogenesis and perichondrial maturation with cartilage development [34] and to cooperate with VEGF in tumor angiogenesis [35]. However, a role for Ihh in the ischemic skeletal muscle has not been demonstrated so far.

### 2.3. Therapeutic Potentials of Exogenous Activation of the Hh Pathway in Ischemic Skeletal Muscle 

Regarding potential therapeutic implications, it has been demonstrated that Shh recombinant protein, injected intraperitoneally in aged mice, augments blood-flow recovery and limb salvage following operatively induced hindlimb ischemia [9]. Also, intramuscular treatment with phShh increases blood flow, capillary density, and arteriole density in old mice in which peripheral circulation of the hindlimb has been disrupted by removal of the common femoral artery [27]. Shh gene therapy also enhances vasculogenesis, by increasing the number of circulating bone marrow (BM)-derived endothelial progenitor cells (EPCs) and improving the contribution of these cells to the process of neovascularization [15]. Combined therapy with Shh gene transfer and BM-derived EPCs is more effective than Shh gene therapy alone in an experimental model of peripheral limb ischemia [36]. The importance of the Hh pathway in the pathophysiology of the EPC compartment has been recently further strengthened by the demonstration that Gli1 and Ptch expression are reduced in the EPCs of streptozotocin (STZ)-induced type 1 diabetic mice and that administration of an Shh pathway receptor agonist (SAG) restores both the number and function of EPCs and increases neovascularization in these mice in response to ischemia [37]. Finally, there are data supporting a potential therapeutic role also for Dhh in the ischemic skeletal muscle. In particular, it has been shown that Dhh supports peripheral nerve survival and maintenance of the pool of nerve-derived proangiogenic factors in the ischemic muscle and that the rescue of Dhh expression by gene therapy in old mice promotes ischemia-induced angiogenesis in the murine skeletal muscle [16]. 

A summary of the results of the studies conducted on the ischemic skeletal muscle is presented in Table 1, while a schematic representation of the mechanisms through which the Hh pathway regulates angiogenesis in the ischemic muscle is presented in Figure 1. 

## 3. Heart

### 3.1. Activation of the Hh Pathway in the Ischemic Heart 

Ischemia results in the upregulation of Hh signaling in the heart, with strong overexpression of Shh and Ptch in the adult mouse during myocardial ischemia [38,39,40]. Although it is likely that, similar to what happens in the skeletal muscle, the main driver of Hh upregulation in the ischemic heart is hypoxia; the literature is particularly poor on this topic, with a study by Bijlsma and coll. that demonstrated that hypoxia stimulates an accumulation of HIF-1α, followed by a slower increase in mature Shh proteins in cultured cardiomyoblast cells [18]. Hwang and coll., as well, have evaluated the role of HIF-1α in a cardiomyoblast cell culture assay under hypoxia, finding that the inhibition of HIF-1α—by means of the HIF-1α inhibitor 2-methoxyestradiol (2ME2)—results in inhibition of the Hh pathway [41]. On the other hand, the same study has also shown that the inhibition of the Hh pathway—by means of cyclopamine—leads to inhibition of HIF-1α in cardiomyoblasts cultured in hypoxic conditions, suggesting the existence of a reciprocal cross-talk between the two pathways [41]. In vivo studies, ideally with knockdown or genetic ablation of HIF-1α in cardiomyocytes, are needed to elucidate the role of HIF-1α in ischemia-induced upregulation of the Hh pathway in the heart. 

### 3.2. Effects of Endogenous Hh in the Ischemic Heart 

There is evidence that the endogenous Hh pathway has a homeostatic role in the ischemic myocardium. In particular, there is a seminal paper by Levine and coll. that demonstrates that myocardial Hh signaling is required for proper cardiac anatomy and function even in the absence of ischemia, as it is responsible for the integrity and maintenance of coronary vessels [42]. Xiao and coll. have reported that, in streptozotocin-induced type 1 diabetic mice, myocardial expression of Shh, Ptch, and Gli1 is significantly decreased, and this is accompanied by cardiac dysfunction [40]. Controversially, Bijlsma and coll. presented data indicating a deleterious effect of endogenous Shh in myocardial ischemia. According to this study, the inhibition of Hh signaling through cyclopamine after the induction of myocardial ischemia in mice would lead to a worse recovery of the left ventricular function, an increase in the number of apoptotic cells, and, counterintuitively, a reduction in left ventricular fibrosis. An immunohistochemical study of the cardiac tissue treated with cyclopamine brought the authors to affirm that endogenous Shh does not induce vascularization after ischemia. These findings are in conflict with most studies published in the literature. For instance, Levine and coll. have shown that the inhibition of the Hh pathway through anti-Shh antibodies leads to a reduction in border zone coronary vessel density and an associated increase in infarct area [42]. Even more convincing evidence was published by Renault and coll., who demonstrated that Gli3 haploinsufficient mice have reduced capillary density and left ventricular ejection fraction after myocardial infarction [14]. Hh-mediated effects in the ischemic heart probably occur through a double mechanism, which includes both enhancement of angiogenesis and direct effects on cardiomyocytes. Indeed, there are convincing data showing that stimulation of angiogenesis takes place in the heart mainly using the same mechanisms that we have described in the skeletal muscle, i.e., production of VEGF and/or other angiogenic growth factors by Hh-responding cells [21,38,43] and no direct effects on ECs by the endogenous Hh signaling [26]. In contrast, it is accepted that Hh is able to exert direct effects on cardiomyocytes. Indeed, a microarray analysis has identified almost 40 genes specifically regulated by Shh in cardiomyocyets, including some in the protein-kinase A (PKA) and purinergic signaling pathways [44].

### 3.3. Therapeutic Potentials of Exogenous Activation of the Hh Pathway in the Ischemic Heart 

Several experimental studies have tested the possibility of enhancing angiogenesis and improving heart function after myocardial ischemia by acting on the Hh pathway. Our group has used intramyocardial gene transfer of naked DNA encoding human Shh (phShh) to enhance neovascularization, reduce fibrosis, and preserve left ventricular function in rabbit and pig models of acute and chronic myocardial ischemia [38]. Ahmed and coll. have overexpressed Shh in rat mesenchymal stem cells (MSCs) [45]. The consequence has been increased expression of angiogenic and pro-survival growth factors in transfected cells, along with improved ability to migrate and form tubes in vitro. When injected in the ischemic myocardium, Shh-overexpressing MSCs lead to an increased density of functionally competent blood vessels and preserved function of the left ventricle. Guo and coll. have overexpressed Shh in cardiac microvascular endothelial cells (CMECs) isolated from rat heart tissues [46]. This has resulted in increased CMEC viability and decreased apoptosis, along with increased production of angiogenic factors, including VEGF, fibroblast growth factor (FGF), and angiopoietins. In a rat model of ischemia-reperfusion, stimulation of an Shh pathway prior to reperfusion has reduced both infarct size and subsequent arrhythmias by preventing ventricular repolarization abnormalities [47]. This is an interesting cardio-protective effect of Shh, which seems to depend on a direct action on cardiomyocytes. Shh has also been reported to promote the angiogenic capacity of BM-EPCs in the heart [38]. Finally, it has been recently reported that engineered MPs carrying Shh (MP-Shh+) may be used to increase the vasculogenic abilities of EPCs and their capacity to produce NO [48]. 

The results of the studies conducted on the ischemic heart are summarized in Table 2, while a schematic representation of the mechanisms through which the Hh pathway regulates angiogenesis in the ischemic heart is presented in Figure 2. 

## 4. Brain

### 4.1. Activation of the Hh Pathway in the Ischemic Brain

A number of data support the concept that Hh signaling is an important regulator of neurogenesis and angiogenesis in the adult brain. For instance, it is known that Shh is required for the maintenance of neural progenitor cells (NPCs) in the adult hippocampus and regulates the proliferation of NPCs after ischemia/hypoxia [49]. Middle cerebral artery occlusion is associated with increased mRNA encoding both Shh and its transcription factor Gli1, already a few hours after induction of brain ischemia [49]. Regarding the molecular mechanisms underlying Hh activation in the ischemic brain, neuroinflammation is a documented trigger [50,51,52]. Shh transcription is induced in astrocytes upon exposure of these cells to inflammatory cytokines and Gli1 is upregulated in the inflammatory peri-ischemic area in the early stage of stroke [51]. Less documented is the role of hypoxia, although it is known that NPCs and neurons increase Shh mRNA levels under hypoxic conditions and that Shh is secreted by activated astrocytes when these cells are incubated under oxygen-glucose deprivation (OGD) conditions [53]. Not only OGD but oxidative stress has also been shown to have the ability to activate the Shh pathway in cultured cortical neurons [54].

### 4.2. Effects of Endogenous Hh in the Ischemic Brain

The importance of endogenous Shh upregulation in the ischemic brain is demonstrated by the fact that, upon induction of experimental cerebral ischemia, inhibition of the Shh pathway results in increased infarct volume, brain water content, and behavioral deficits [55]. The endogenous Hh pathway appears to be important also for the beneficial effects induced by bone marrow stromal cells (MSCs) on neurologic recovery after middle cerebral artery occlusion in mice [56]. Indeed, treatment with MSCs significantly enhances functional recovery after middle cerebral artery occlusion, concurrent with increases of synaptophysin, synapse density, and myelinated axons along the ischemic boundary zone. However, these benefits are significantly reduced by blockage of the endogenous Shh pathway [56]. Also, the beneficial effects displayed by cerebrolysin—a mixture of neurotrophic peptides—in experimental neurodegenerative diseases and stroke seem to depend on proper activity of the endogenous Hh pathway [57]. Indeed, cerebrolysin significantly increases neural progenitor cell proliferation and differentiation into neurons and myelinating oligodendrocytes in rats subjected to embolic stroke. However, blockage of the Shh signaling pathway with a pharmacological smoothened inhibitor, cyclopamine, abolishes cerebrolysin-induced in vitro neurogenesis and oligodendrogenesis. Moreover, profound neurological function improvements were observed in rats treated with cerebrolysin from week three to week five after stroke onset. However, in vivo inhibition of the Shh pathway with cyclopamine completely reverses the effects of cerebrolysin on functional recovery. At the mechanistic level, there are data indicating that activation of endogenous Shh signaling protects neurons from H_2_O_2_-induced apoptosis and increases cell viability [54]. Regarding a possible role of the endogenous Hh pathway on angiogenesis in the ischemic brain, it has been shown that Shh produced by astrocytes (when these cells are incubated in under OGD condition) enhances proliferation, migration, and tube formation of brain microvascular endothelial cells (BMECs) [54].

### 4.3. Therapeutic Potentials of Exogenous Activation of the Hh Pathway in the Ischemic Brain 

In H_2_O_2_-treated neurons, exogenous Shh increases the activity of Superoxide dismutase (SOD) and Glutathione peroxidase (GSH-PX) and decreases the production of Malondialdehyde (MDA). It also promotes the expression of the anti-apoptotic gene Bcl-2 and inhibits expression of pro-apoptotic gene Bax. Activation of the Shh signal also leads to an upregulated expression of both neurotrophic and vascular factors, such as VEGF and brain-derived neurotrophic factor (BDNF) [54,58]. Intracerebroventricular injections of Shh in rats immediately after permanent middle cerebral artery occlusion reduces infarct volume, leads to improved microvascular density and neuron survival in the ischemic boundary zone, and improves neurological scores [21]. These beneficial effects occur along with enhanced VEGF expression [21]. Stimulation of the Shh pathway appears to be beneficial even if performed not immediately after stroke. Indeed, delayed treatment (one month after stroke) with an Shh agonist in mice resulted in enhanced functional recovery both in locomotor and cognitive function [59]. Furthermore, using a genetically inducible neural stem cell-specific reporter mouse line (nestin-CreERT2-R26R-YFP), which allows the labelling and tracking of neural stem cell proliferation, survival, and differentiation in ischemic brain, it was shown that post-stroke treatment with an Shh pathway agonist increases survival of newly born cells derived from both subventricular and subgranular zones and neurons in the ischemic brain [59]. Huang and coll. have tested whether the epidural application of the biologically active N-terminal fragment of Shh (Shh-N) may reduce the extent of ischemic brain injury in rats exposed to a 60 min episode of middle cerebral artery occlusion. Shh-N was applied topically by placing a fibrin glue over the peri-infarct cortex [60]. They found that such treatment substantially reduced infarct volumes after seven days of reperfusion. It also improved behavioral outcomes as assessed by global neurological functions, rotarod test, and grasping power test. Furthermore, Shh-N attenuated the extents of protein oxidation, lipid peroxidation, and apoptosis induced by focal ischemia/reperfusion. Immunohistochemical staining revealed that Shh-N enhanced post-ischemic angiogenesis stimulated the proliferation of nestin-positive neural progenitor cells (NPCs) and suppressed astrocytosis.

The results of the studies conducted on the ischemic brain are summarized in Table 3, while a schematic representation of the mechanisms through which the Hh pathway regulates angiogenesis in the ischemic brain is presented in Figure 3. 

## 5. Other Ischemic Tissues

In addition to the skeletal muscle, heart, and brain, there are some organs, such as liver, lung, and kidney, in which it has been brought to evidence that the activation of the Hh pathway may be involved in pathologic angiogenesis, with an effect on fibrogenesis and whole diseased organs fibrotic degeneration [61].

### 5.1. Liver

The contribution of Shh signaling to fibrogenesis has been particularly studied in the liver. Hypoxia-induced pathological angiogenesis is an important phenomenon closely correlated to the disruption of the hepatic architecture occurring in chronic liver diseases. Feng and coll. elegantly demonstrated the pro-angiogenic and pro-fibrinogenic role of the canonical Hh pathway in a model of liver fibrosis in rats. First, they observed high expression of Shh and other hypoxia-associated proteins, such as VEGF and HIF-1α, in the setting of hepatic fibrosis, consistent with the existence of a hypoxic micro-environment. Then, they found that treatment with Gli1 antagonist (GANT)-58 results in reduced the expression of VEGF and Ang-1 and inhibits hepatic stellate cell (HSC)-mediated tubulogenesis. These actions appear to be mediated by the interaction of Gli1 with the *HSP90* gene, involved in the stabilization of HIF-1α in an oxygen-independent manner. This study brings to evidence that Shh-induced angiogenesis in liver hypoxia may be more associated with hepatic fibrosis than to liver regeneration [62]. Pratap and coll. demonstrated that inhibition of Shh limits ischemia-reperfusion injury (IRI) in cholestatic rat livers. Preconditioning bile duct ligated rats with cyclopamine before subjecting them to IRI lead to decreased levels of serum aspartate aminotransferase (AST), alanine aminotransferase (ALT), and bilirubin levels, reduced liver damage at the histological level, reduced levels of neutrophil infiltration and expression of proinflammatory cytokines, and inhibited HSC activation and cholangiocyte proliferation [63]. Conversely, Tuncer and coll. have shown that the upregulation of Shh through treatment with L-Arginine (L-Arg) in a rat hepatic IRI model reduces indices of hepatocellular necrosis and leads to a better histopathological score when compared to untreated ischemic livers. However, it is not possible to ascribe the effects obtained in the study solely to Shh, due to the concomitant presence of nitric oxide (NO) released by L-Arg [64]. 

### 5.2. Lung

Shh signaling in the embryonic respiratory epithelium appears to have a crucial role in the branching morphogenesis of the lung, and the expression of Ptch by lung mesenchymal cells is vital for normal lung formation [65]. In the adult lung, the role of Shh has been mainly studied in relation to the etiology of chronic diseases, such as asthma [66], chronic obstructive pulmonary disease (COPD) [67], and idiopathic pulmonary fibrosis [68], and lung carcinogenesis [69]. Very few papers, however, are available on the role of Hh in the hypoxic pulmonary tissue, and none on the lung in the setting of ischemia. Wang et al. demonstrated in vitro that hypoxia markedly activates the Shh pathway in human pulmonary arterial smooth muscle cells (HPASMCs) and that the proliferation of these cells in response to the ischemic injury is mediated at least in part by the Shh pathway [70]. Al Ghouleh et al. investigated the molecular mechanisms responsible for aberrant vascular remodeling occurring in pulmonary arterial hypertension (PAH) patients, demonstrating an up-regulation of nicotinamide adenine dinucleotide phosphate (NADPH) oxidase 1 (Nox1), an increase in reactive oxygen species (ROS) production and expression of bone morphogenetic protein (BMP) antagonist Gremlin1 (Grem1) in resistance vessels from PAH patients compared with non-PAH patients [71]. In human pulmonary arterial endothelial cells (HPAECs), hypoxia induced Nox1 subunit expression, assembly, and oxidase activity leading to an elevation in Sonic hedgehog and Grem1 expression. The authors stated that these data support a Nox1-Shh-Grem1 signaling axis in pulmonary vascular endothelium, likely contributing to the pathophysiological endothelial proliferation underlying PAH [71].

### 5.3. Kidney 

Few studies analyzed the effect of Shh on the ischemic kidney. Some attention has been paid to the pro- or anti-fibrotic activity of Shh in renal IRI. Ozturk et al. studied the expression of Shh in murine models of IRI after treatment with L-Arg, a precursor of NO [72]. They showed that treatment with L-Arg produced a significant overexpression of Shh in tubular epithelial cells, compared with the sham-control and the IR/untreated group, and reduced the renal dysfunction associated with IRI of the kidney [72]. Guanqun et al. studied the role of Shh-Gli1 signaling in kidney regeneration after renal IRI [73]. They showed that IRI activates Shh-Gli1 signaling and is furthermore responsible for the up-regulation of the ATP-binding cassette, subfamily G, member 2 transporter (ABCG2), an essential element for kidney regeneration after renal IRI [73]. Similarly, Meng et al. first confirmed that the expression of Shh in ischemic kidneys is significantly higher than in non-ischemic kidneys [74]. Afterwards, based on the evidence that polydatin, a glucoside of resveratrol extracted by the dried roots of *Polygonum Cuspidatum Sieb.*, exhibits beneficial effects in ischemic organs such as heart, brain, lungs, and kidneys, demonstrated that blocking the Shh pathway (through cyclopamine or the Shh antibody 5E1) markedly suppressed the positive effects of polydatin both in ischemic kidneys in unilateral renal IRI mice in vivo and in renal tubular cells under OGD in vitro [74]. Metabolic derangements, such as hyperglycemia, are known to impair normal wound healing through a hypothesized mechanism involving the persistent activation of profibrotic signaling pathways, such as transforming growth factor (TGF)-β. Indeed, recovery from transient kidney damage is poorer in the presence of underlying diseases such as diabetes, leading to a higher incidence of transition from acute kidney injury (AKI) to chronic kidney injury (CKD) in diabetic patients. Despite the recognized role of Shh in leading to kidney fibrosis following AKI, its relationship with hyperglycemia is unclear. The results of a study by Dong-Jin et al. suggest that diabetes induces persistent activation of Shh signaling and indicate that hyperglycemia itself could be responsible for the activation of Shh during CKD progression after kidney IRI in diabetic mice [75]. Moreover, in vitro experiments on renal tubular cells show that hyperglycemia augments the effect of Shh on their profibrogenic phenotype change, thus contributing to the hypothesis that Shh may have a role in the development of CKD in diabetic patients [75]. 

## 6. Canonical Versus Non-Canonical Pathways

Canonically, Hh signaling occurs through the binding of Hh proteins to the twelve-transmembrane domain receptor Ptch. Binding of a Hh ligand to Ptch results in Ptch internalization and abolishment of its inhibitory activity on the seven-transmembrane protein Smoothened (Smo). Smo is subsequently phosphorylated at its intracellular C-terminus and undergoes reciprocal translocation to the plasma membrane of non-motile cilia. Through an unknown mechanism, ciliary Smo inhibits the downstream proteolytic processing of GLI2 and GLI3, leading to the production of full-length activated Gli proteins (GliA). GliA enter the nucleus and stimulate transcription of the ubiquitous Hh target genes, GLI1, PTCH, and HHIP (Hh interacting protein). When the pathway is off, Gli2 and Gli3 proteins are converted into truncated repressor forms, named GliR (Gli repressors), which inhibit target gene transcription. When the pathway is on, GliR production is blocked, and Gli2 and Gli3 proteins are converted into GliA [76]. The balance between GliR and GliA is of great relevance. Indeed, changes in the GliR/GliA ratio can lead to developmental defects in humans, demonstrating that the precise level of Gli activity is critical for the sophisticated patterning events regulated by Hh signaling during development [77]. In addition to this canonical pathway, there are cellular and tissue responses to Hh ligands that occur through non-canonical pathways, which mainly utilize one of the three following mechanisms: (i) core components of Hh signaling interact with each other in a non-contiguous or atypical way; (ii) components of Hh signaling directly interact with constituents of other molecular pathways; (iii) Hh signaling occurs via activation of Smo in an Gli-independent way, or via activation of Ptch in an Smo-independent way [24,76,78].

A limitation of the vast majority of the studies listed in this review is that they have not evaluated which pathway is acting in what situations. One exception is the study published by Renault and coll. in 2009, in which it is shown that Gli3-deficient mice have reduced capillary density after induction of hindlimb ischemia [17]. They also have reduced left ventricular ejection fraction after myocardial infarction [17]. This indicates that Gli is important in myocardial and muscular response to ischemia. Another exception is a study published by Gupta and coll. in 2018, in which the ablation of the canonical pathway in ECs does not affect angiogenic response to ischemia [26]. This finding supports the concept that, in ischemic situations, Hh proteins do not exert a direct effect on ECs, at least through the canonical pathway. They instead seem to act through non-canonical pathways, as suggested by the evidence that Hh proteins may stimulate tubulogenesis in ECs by RhoA via Smo and Gi proteins [24,79,80]. Also, there is evidence that Shh binding to RhoA GTPase via a non-canonical Smo-dependent pathway leads to the maturation of EPCs to ECs (a crucial step in angiogenesis and vasculogenesis) [24,25,79,80]. Another study that provides some information on the pathway(s) used by Hh to exert its effects in the ischemic heart is that published by Carbe and coll. in 2014 [44]. In this study, neonatal rat ventricular cardiomyocytes infected with an adenovirus encoding GiCT, a peptide that impairs receptor-Gi protein coupling, showed reduced activation of Hh targets. In vitro data were confirmed in transgenic mice with cardiomyocyte-inducible GiCT expression [44]. Transgenic GiCT mice showed a specific reduction of Gli1 expression in the heart under basal conditions and failed to upregulate the Hh pathway upon ischemia and reperfusion injury [44]. Of note, none of the studies dealing with the Hh pathway in the ischemic brain contains relevant information on which pathways are involved in the various pathophysiological situations. Indeed, they generally use cyclopamine to inhibit the Hh pathway or SAG (a Sonic hedgehog agonist) to stimulate the pathway. However, these methodologies do not help to distinguish between pathways in a rigorous manner. 

## 7. Concluding Remarks

Once believed to be important only during embryonic development, the Hh signaling pathway is now recognized as a crucial player in many physiological and pathological conditions in post-natal life. Its role during ischemia-induced angiogenesis and tissue repair has been extensively investigated in the last two decades and seems to be particularly important in organs such as the heart, the brain, and the skeletal muscle. Although the precise molecular and cellular mechanisms underlying Hh-mediated angiogenesis still require full elucidation, there is substantial evidence supporting the concept that the endogenous Hh signaling pathway has the ability to orchestrate multiple aspects of the angiogenic response to ischemia, mainly through the production of angiogenic growth factors from interstitial cells and fibroblasts and the regulation of EPC activity. Nevertheless, there is evidence that many activities of Hh proteins in the ischemic heart and brain occur through direct stimulation of cardiomyocytes and neural cells. Based on the concept that the endogenous Hh pathway is important for efficient angiogenic response to ischemia, several studies have focused on the use of agonists of the Hh pathway to stimulate therapeutic angiogenesis. Successful results have been achieved in experimental models of myocardial infarction, ischemic stroke, and peripheral limb ischemia. However, at the moment, there is a complete lack of data on human subjects, and doubts remain on the actual possibility of manipulating the Hh pathway in a safe manner in humans. In this context, it is worth pointing out that Shh has been implicated in the pathogenesis of several types of cancer and that anti-Hh strategies are currently used, or are under investigation, for the treatment of many neoplastic diseases, including basal cell carcinoma, acute promyelocytic leukemia, medulloblastoma, small cell lung cancer, pancreatic cancer, intracranial meningioma, recurrent glioblastoma, prostate cancer, renal cell carcinoma, and colon cancer. Based on this, it is unseemly that none of the experimental studies performed so far made an effort to evaluate whether stimulation of angiogenesis by Shh agonists in the ischemic heart, brain, and skeletal muscle is safe or may have dismal consequences. Studies testing the safety of exogenous administration of Hh agonists are therefore mandatory in order to understand whether the large amount of experimental data produced so far has translational significance and potential clinical implications.

## Figures and Tables

**Figure 1 ijms-20-05270-f001:**
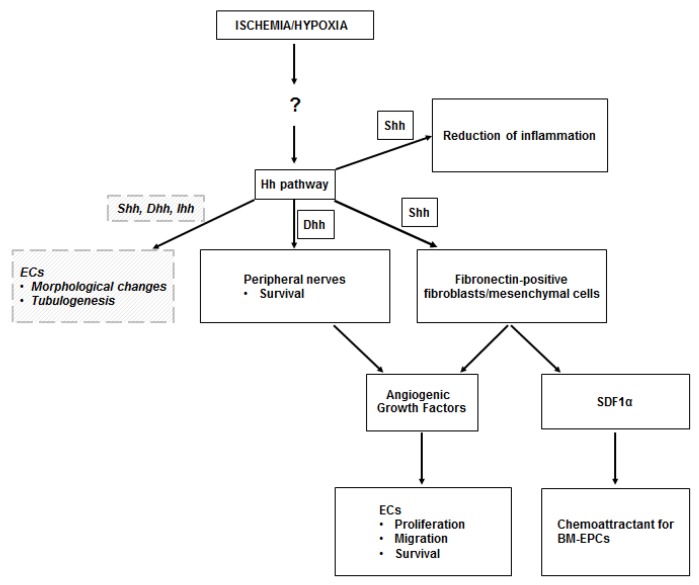
Hh in the ischemic skeletal muscle. The upregulation of the Hedgehog (Hh) pathway observed in the ischemic skeletal muscle occurs through unknown mechanisms, perhaps mediated by the Hypoxia Inducible Factor-1α (HIF-1α). Sonic hedgehog (Shh) protein acts on fibroblasts, which respond by producing angiogenic growth factors (including VEGF, HGF, PDGF-BB, IGF), which stimulate proliferation, migration, and survival of endothelial cells (ECs). Shh-responding fibroblasts also produce stromal cell-derived factor 1α (SDF1α), which is responsible for the recruitment of bone marrow-derived endothelial progenitor cells (BM-EPCs) into the ischemic site. Angiogenic growth factors are also produced by peripheral nerves in the ischemic muscle, upon Dhh modulation. There is also evidence that Shh limits the inflammatory response to ischemia in the skeletal muscle. Finally, there is in vitro evidence that ECs may respond to Hh proteins in a direct manner. Such direct stimulation induces EC morphological changes and supports EC tubulogenesis. Dashed lines and italic text represent in vitro data.

**Figure 2 ijms-20-05270-f002:**
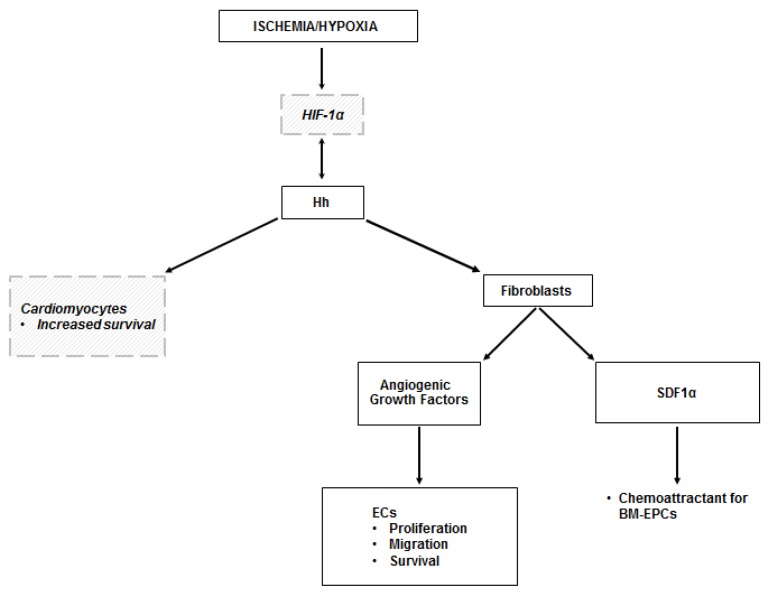
Hh in the ischemic heart. The mechanisms leading to the activation of the Hh pathway in the ischemic heart are not fully elucidated. It is likely that they depend on HIF-1α, but this has been demonstrated only in vitro. There is also evidence of a reciprocal activation of HIF-1α by Shh. Hh-dependent angiogenesis in the ischemic heart occurs indirectly, as it is mediated by the production of angiogenic growth factors by myocardial fibroblasts in response to Hh stimulation. Hh-responding fibroblasts also produce SDF1α, which is responsible for the recruitment of BM-EPCs in the ischemic area. There are in vitro data showing that Shh has the ability to regulate the transcriptional response of cardiomyocytes, with increased expression of genes involved in cell survival. Dashed lines and italic text represent in vitro data.

**Figure 3 ijms-20-05270-f003:**
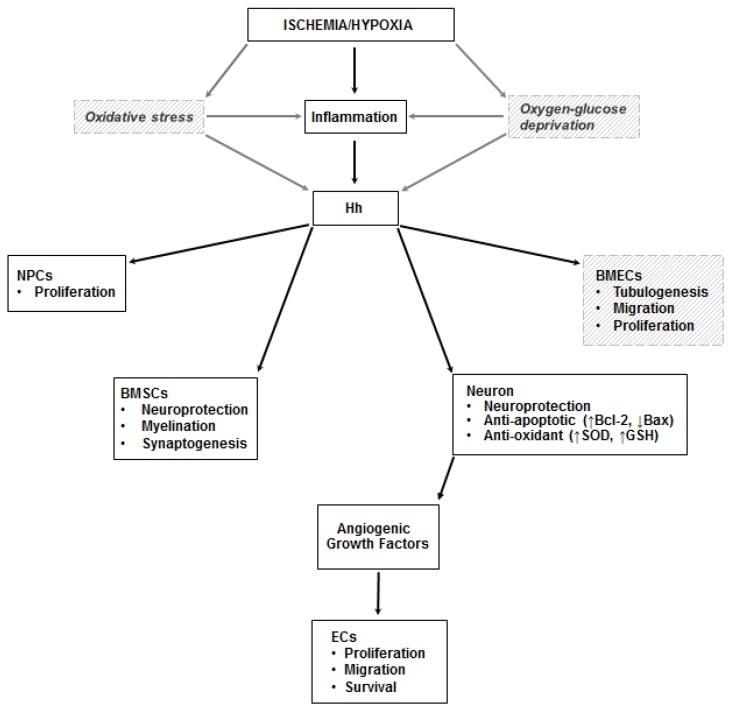
Hh in the ischemic brain. Inflammation is a documented trigger of the Hh pathway in the ischemic brain. Less compelling (and only in vitro) evidence exists on the role of oxidative stress and oxygen-glucose deprivation on the stimulation of the pathway. In neurons, Shh exerts its neuroprotective effects by increasing the activity of superoxide dismutase (SOD) and glutathione peroxidase (GSH-PX), and by promoting the production of Bcl-2 (the black arrow pointing up indicates upregulation) and inhibiting expression of Bax (the black arrow pointing down indicates downregulation), thus exerting anti-oxidant and anti-apoptotic effects. It also increases the proliferation of neural progenitor cells (NPCs), stimulates myelination and synaptogenesis in bone marrow stromal cells (BMSCs), and promotes tubulogenesis, migration, and proliferation of brain microvascular endothelial cells (BMECs). Angiogenic growth factors are produced by neurons stimulated by Shh. Dashed lines and italic text represent in vitro data.

**Table 1 ijms-20-05270-t001:** Hedgehog in the ischemic skeletal muscle.

Result	First Author	Journal and Year of Publication	Ref.
Hindlimb ischemia induces Shh and Ptch upregulation in mice	Pola	Circulation, 2003	[13]
Hh inhibition with 5E1 antibody decreases upregulation of VEGF in response to ischemia	Pola	Circulation, 2003	[13]
Gli3-haploinsufficient (Gli3+/−) mice have reduced capillary density after induction of hindlimb ischemia	Renault	Circulation Res, 2009	[17]
None of the Hh ligands is able to induce Gli-target genes in endothelial cells Shh, Dhh, and Ihh proteins promote endothelial cell tubulogenesis and cytoskeletal rearrangement through non-canonical Hh pathways	Chinchilla	Cell Cycle, 2010	[24]
Endothelial-specific Smo knockout (eSmo^null^) mice do not have reduced angiogenic response to ischemia	Gupta	Lab Invest, 2018	[26]
Inhibition of the endogenous Shh pathway in Shh inducible knock out (Shh iKO) mice does not impair angiogenic response to ischemia Inflammation and macrophage recruitment are increased in the ischemic skeletal muscle when the endogenous Shh pathway is inhibited in Shh iKO mice	Caradu	Cardiovasc Res, 2018	[30]
Intraperitoneal injection of Shh recombinant protein enhances angiogenic response to ischemia in aged mice	Pola	Nature Med, 2001	[9]
Intramuscular treatment with a plasmid encoding the Shh human gene (phShh) enhances angiogenic response to ischemia in aged mice	Straface	J Cell Mol Med, 2009	[27]
Intramuscular treatment with phShh enhances skeletal muscle regeneration in aged mice	Piccioni	J Gerontol A Biol Sci Med Sci, 2014	[28]
Intramuscular treatment with phShh increases the number of bone marrow (BM)-derived endothelial progenitor cells (EPCs) in response to ischemia in mice	Palladino	Molecular Therapy, 2011	[15]
Combined therapy with phShh and BM-derived EPCs enhances angiogenesis and myogenesis in the murine ischemic skeletal muscle	Palladino	J Vasc Res, 2012	[36]
Treatment with an Shh receptor agonist (SAG) restores number and function of EPCs in diabetic mice and increases neovascularization in response to ischemia	Qin	Mol Cell Endocrinol, 2016	[37]
Dhh gene therapy enhances angiogenesis in the ischemic muscle of aged mice through promotion of peripheral nerve survival	Renault	Circulation Res, 2013	[16]
Treatment with a Dhh-signaling agonist (SAG) improves endothelial function in diabetic mice without promoting angiogenesis	Caradu	Circulation Res, 2018	[31]
Microparticles carrying Shh are increased in humans with peripheral artery disease	Giarretta	Int J Mol Sci, 2018	[29]

**Table 2 ijms-20-05270-t002:** Hedgehog in the ischemic heart.

Result	First Author	Journal and Year of Publication	Ref.
Ischemia induces upregulation of Shh protein in the adult mouse heart	Kusano Bijlsma Xiao	Nature Med, 2005 Exp Biol Med, 2008 Cardiovasc Res, 2012	[38] [39] [40]
HIF-1α is responsible for Hh activation in hypoxic cardiomyoblasts	Hwang Bijlsma	Mol Cell Biochem, 2008 J C Mol Med, 2009	[41] [18]
Endogenous Hh expression does not induce vascularization, but rather increases apoptosis and reduces fibrosis in murine ischemic myocardium	Bijlsma	Exp Biol Med, 2008	[39]
Hh signaling is required for the maintenance of the adult coronary vasculature Endogenous Hh signaling reduces infarct area and increases border zone coronary vessel density after myocardial infarction in mice	Lavine	J Clin Invest, 2008	[42]
Gli3-haploinsufficient (Gli3+/-) mice have reduced capillary density and left ventricular ejection fraction after myocardial infarction	Renault	Circulation Res, 2009	[17]
Upregulation of Shh, Ptch, and Gli1 is impaired in the ischemic myocardium of diabetic mice	Xiao	Cardiovasc Res, 2012	[40]
Canonical Shh signaling regulates gene transcription in cardiomyocytes	Carbe	Am J Physiol Heart Circ Physiol, 2014	[44]
Treatment with an Shh pathway agonist (SAG) enhances capillary density, reduces infarct size, and improves cardiac function in diabetic mice with myocardial infarction	Xiao	Cardiovasc Res, 2012	[40]
Intramyocardial gene transfer of phShh induces production of angiogenic growth factors, enhances neovascularization, reduces fibrosis, and preserves left ventricular function in rabbit and pig models of acute and chronic myocardial ischemia	Kusano	Nature Med, 2005	[38]
Injection in the rat ischemic myocardium of Shh-overexpressing mesenchymal stem cells (MSCs) increases blood vessel density and preserves function of the left ventricle	Ahmed	PLoS One, 2010	[45]
Intramyocardial gene transfer of phShh increases the number of bone marrow BM-EPCs and their differentiation into endothelial cells in the myocardium	Kusano	Nature Med, 2005	[38]
In a rat model of ischemia-reperfusion, stimulation of Shh pathway (by a recombinant peptide or microparticles harboring Shh) reduces infarct size and arrhythmias	Paulis	Sci Rep, 2015	[47]

**Table 3 ijms-20-05270-t003:** Hedgehog in the ischemic brain.

Result	First Author	Journal and Year of Publication	Ref.
Neuroinflammation triggers Hh activation in the ischemic brain	Gu Amankulor Lalancette-Hebert	Neurochem Res, 2015 J Neurosci, 2009 J Neurosci, 2007	[50] [51] [52]
Middle cerebral artery occlusion (MCAO) induces upregulation of Shh and Gli1 in the mouse brain Inhibition of the Hh pathway (with cyclopamine) suppresses ischemia-induced proliferation of subgranular neural progenitor cells (NPCs) in mice	Sims	Stroke, 2009	[49]
Inhibition of the Hh pathway (with cyclopamine) after permanent MCAO increases infarct volume, brain water content, and behavioral deficits in rats	Ji	Neurosci Lett, 2012	[55]
Inhibition of the Hh pathway (with cyclopamine) reduces the beneficial effects of treatment with bone marrow stromal cells (MSCs) after MCAO	Ding	J Cereb Blood Flow Metab, 2013	[56]
Inhibition of the Hh pathway (with cyclopamine) suppresses the beneficial effects of cerebrolysin in models of stroke	Zhang	Stroke, 2013	[57]
Intracerebroventricular injections of Shh protein in rats immediately after permanent MCAO reduces infarct volume, enhances microvascular density and neuron survival in the ischemic boundary zone, and improves neurological scores	Chen	Neuroscience, 2017	[21]
Delayed treatment (1 month after stroke) with an Shh agonist enhances functional recovery in locomotor and cognitive function in mice	Jin	Stroke, 2017	[59]
Post-stroke treatment with an Shh agonist (SAG) increases surviving newly born cells derived from the subventricular and subgranular zone and neurons in the mouse ischemic brain	Jin	Stroke, 2017	[59]
Epidural application of biologically active N-terminal fragment of Shh enhances angiogenesis, stimulates proliferation of NPCs, suppresses astrocytosis, reduces infarct volume, and improves behavioral outcomes in rats exposed to a 60-min episode of MCAO	Huang	Exp Neurol, 2013	[60]

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
