# Peer review of "The Hedgehog Signaling Pathway in Ischemic Tissues"

_ijms, 2019, doi:10.3390/ijms20215270_

Round 1
Reviewer 1 Report
1. “I would like to know the big picture, how does the hedgehog pathway regulates angiogenesis and ischemic lesion overall, and in that sense I would like to know what happens in the ischemic lung, kidney and liver.”
2. “Are there differences in angiogenic response and ischemic tissue repair in the different organs? For instance, in the skeletal muscle the hedgehog pathway seems to have only indirect effect on angiogenesis through the production of cytokines, whereas in the brain it directly promotes proliferation, migration and tube formation of BMECs. I did not understand by the authors’ review if there is a direct effect in the heart also.”
3. “It would be interesting to have a simplified description of the physiological response to ischemia in the different tissues and then to point out what is the specific role of hedgehog in that pathway. For example, I believe that the effect of hedgehog, for instance in the ischemic muscle is more related to the enhancement of the wound-healing response, rather than an effect on angiogenesis.” ................. “It would be nice to have a diagram explaining the hedgehog response and another one explaining the effects of hedgehog on ischemic response.”
4. “Could the authors present human data in the different tissues?”
5. “Regarding the chapter on heart, the fact that inhibition of the hedgehog pathway does not induce ECG changes implies lack of effect on the conduction system, but does not imply a lack of effect on the heart.”
6. “Hedgehog overexpression in the liver promotes an exaggerated wound-healing response inducing liver and kidney fibrosis, however, the authors stated that overexpression of hedgehog in the ischemic heart reduces fibrosis. Could the authors comment on that? Could it be because in the heart the authors performed an acute rather than chronic hedgehog overexpression?”
7. “When the authors state that lack of hedgehog increases infarct and neurologic deficits, it should be specified that those findings were in rats.”
Author Response
- We had opted to focus on skeletal muscle, heart muscle, and brain since those are the tissues most studied in regard to their response to Hedgehog (Hh) in ischemic conditions. However, the manuscript has now been edited to include the few data from the literature that deal with the effects of Hh in the ischemic lung, kidney, and liver (pp. 14-15).
- We incorporated subchapters in the revised manuscript, to better define the mechanisms of action of the hedgehog pathway in the various organs and tissues. Also, we made a distinction between the possible mechanisms of action of the endogenous hedgehog pathway and those stimulated by the exogenous activation of the pathway in ischemic muscle, heart, and brain. Other organs were not included due to the paucity of literature data. These additions are on pages 2-3, 6-7, 10-11.
- As requested by the Reviewer, we have included three new figures (Fig. 1, 2, and 3) that schematically describe how the Hh pathway is activated in response to ischemia in the skeletal muscle, heart, and brain and how it regulates the response to ischemia in these tissues. Please note that these are necessarily “simplified” descriptions of the way Hh interacts with the intricate and widespread pathway networks involved in the physiological response to ischemia. Indeed, these pathways are complex, in part still unknown, and different from organ to organ. Nonetheless, our schematic diagrams help understanding the effects of Hh on ischemic response.
- Of all the manuscripts listed in our review, only that by Giarretta and coll. includes human data (detection of Shh-carrying microparticles in the blood of subjects affected by peripheral artery disease) [ref. 29].
- We agree with the Reviewer and consider the above-mentioned paper not relevant for the purposes of this review. Thus, it is not mentioned in the revised version of the manuscript.
- Controversies exist on the role of Hh in the liver. It is indeed reported that Shh upregulation in the ischemic liver is associated with fibrosis, rather than liver regeneration. However, there are also studies suggesting that the upregulation of Shh in the ischemic liver leads to reduced indices of hepatocellular necrosis and a better histopathological score when compared to the untreated ischemic liver. Important links between Shh and fibrosis have been described also in the lung, with a possible role for Shh signaling in idiopathic pulmonary fibrosis (IPF). For instance, it has been recently published a study that demonstrates that mesenchymal stem cells residing in the lung respond to stimulation by recombinant Shh acquiring a myofibroblast phenotype, while inhibition of the Shh signaling prevents the transformation of these cells into myofibroblasts and ameliorates pulmonary fibrotic lesions in an experimental model of IPF. Things seem instead to be different in the ischemic heart, where a large body of evidence supports the ability of Shh to reduce fibrosis. The Reviewer proposes that these differences might depend on the fact that Shh was acutely, and not chronically, overexpressed in the ischemic heart. However, there is evidence that Gli3-haploinsufficient mice, in which the hedgehog pathway is chronically downregulated, have increased fibrosis after myocardial ischemia. This argues against the hypothesis of the Reviewer. All these issues are discussed in detail in the revised version of the manuscript (pp. 7 and 14-15).
- This specification has been made in the revised version of the manuscript (p. 11).
Reviewer 2 Report
“The authors quite correctly point out in their second paragraph that there are several downstream pathways by which Hh may act. These are broadly divided into canonical versus non-canonical .The non-canonical pathways are complex and diverse including regulation of apoptosis (CDON etc), cell migration, calcium flux and metabolic effects. These differences may not be important to define if therapeutic intervention is aimed at the ligand rather than the downstream effector responses. However if the downstream pathways are a potential target then it is absolutely necessary that they be defined for each process .This review reflects the confusion in this area rather than leading to clarification. Unfortunately the review does not systematically examine the data carefully enough to define which pathway is acting in what situations. Furthermore the use of Gli as a specific readout of Hh ligand response is unacceptable as Gli is induced by several factors other than Hh ligands. The Hh field is full of confusion because of the lack of rigor in defining carefully what downstream pathway(s) is involved and when. Such experiments require detection of primary cilia, analysis of CDON etc , Smo ablation (not by cyclopamide) etc … Unfortunately many papers do not explicitly clarify these issues ............ For this review to be useful it would need to clarify what pathways are involved in what situations .If that is unable to be defined then authors could outline what experimentation is required to define the exact downstream pathways involved.”
Author Response
We agree on the fact that confusion exists in this field and are grateful to the Reviewer for raising this issue. In the revised manuscript, we have added a new paragraph entitled “Canonical versus non-canonical pathways” (pp. 15-16). In this paragraph, which deals exclusively with this issue, we point out that the vast majority of the studies listed in this review have the limitation that a distinction between canonical and non-canonical pathways has not been made. There are a few exceptions (Renault and coll. 2009, Gupta and coll. 2018, Carbe and coll. 2014), which are discussed in detail.
Reviewer 3 Report
1. “In page 2, line 47, the authors introduced Gli repressor (GliR); however, the explanation about GliR was not enough. They need to describe the general information about GliR, for examples, 1) how GliR is generated; 2) which Gli family member mainly functions as GliR, and so on.
2. In page 2, lines 92 to 93, the authors described that “further studies are needed to fully understand the molecular mechanisms responsible for Hh regulated angiogenesis”. In page 9 lines 257 to 259, they also described that “further studies are needed to fully understand the translational significance and potential clinical implications of these findings.” The authors need to describe what kind of studies and experiments are needed to achieve these purposes and discuss the future perspectives.
3. “In page 2 and 3, from lines 94 to 100. The authors introduced the controversial results; however, they did not discuss what are the causes of the discrepancies. The authors need to discuss this point.”
4. “In page 3 line 115; page 4, line 138; page 5, lines 165 to 166; and page 7, lines 226 to 227, because there are several hedgehog pathway agonists and inhibitors commercially available, it will be informative to readers if the authors describe which reagents were used in each studies.”
Author Response
- The revised version of the manuscript contains information on how GliR are generated and which Gli proteins mainly function as GliR (p. 15).
- In the revised version of the manuscript, we underline the point that safety studies are needed, in order to understand the potential clinical implications of the many experimental findings produced so far (pp. 16-17).
- This point is extensively discussed in the revised version of the manuscript (p. 3).
- In the revised manuscript, all the tables include information on the specific agonists and inhibitors that were used in any single study.
Round 2
Reviewer 2 Report
The authors have introduced the final paragraph which summarise the limitations of current studies
Reviewer 3 Report
The authors revised the manuscript properly. The manuscript was improved a lot.